# Neuroprognostication value of serum neurofilament light chain for out-of-hospital cardiac arrest: A systematic review and meta-analysis

Yu Fu [1,2], Xiao-Tian Fan [1,2], Hui Li [1,3], Ran Zhang [3], Ding-Ding Zhang [4‡], Hao Jiang [5‡], Zhi-Guo Chen [2]*, Jiang-Tao Zhang [3]*

**1** Graduate School of Chengde Medical University, Chengde, He Bei Provence, China, **2** Department of Emergency, Chengde Central Hospital, Chengde, He Bei Provence, China, **3** Department of Neurology, Chengde Central Hospital, Chengde, He Bei Provence, China, **4** Medical Research Center, Peking Union Medical College Hospital, Beijing, China, **5** Medical Research Center, Chengde Central Hospital, Chengde, He Bei Provence, China

☯ These authors contributed equally to this work.
‡ DDZ and HJ also contributed equally to this work.
* chenzhiguo1210@126.com (ZGC); zhangjiangtaocd@gmail.com (JTZ)

**Data Availability Statement:** All relevant data are within the paper and its Supporting information files.

## Abstract

### Background

Neurofilament light chain (NfL) is a novel biomarker for the assessment of neurological function after cardiac arrest (CA). Although meta-analysis has confirmed its predictive value, it has not conducted a more detailed analysis of its research. We conducted a meta-analysis to evaluate the relationship between serum NfL level and neurological prognosis in patients with spontaneous circulation recovery after CA, and subgroup analysis was conducted according to sample collection time, time to assess neurological function, study design, whether TTM was received, the method of specimen determination, and the presence of neurological disease in patients. To analyze the influence of these factors on the predictive value of serum NfL.

### Methods

Published Cochrane reviews and an updated, extended search of MEDLINE, Cochrane Library, Embase, Scopus, ClinicalKey, CINAHL, and Web of Science for relevant studies until March 2022 were assessed through inclusion and exclusion criteria. The standard mean difference and 95% confidence interval were calculated using the random-effects model or fixed-effects model to assess the association between one variable factor NfL level and the outcome of CA patients. Subgroup analysis according to sample collection time was performed. The prognosis analysis and publication bias were also assessed using Egger's and Begg's tests.

### Results

Among 1209 related articles for screening, 6 studies (1360 patients) met the inclusion criteria and were selected for meta-analysis. The level of serum NfL in the good

**Funding:** The Science and Technology Project of Hebei Province (Approval number :20377764D).

**Competing interests:** The authors have declared that no competing interests exist.

prognosis group (CPC1-2, CPC: cerebral performance category score) was significantly lower than that in the poor prognosis group (CPC3-5)SMD(standardized mean difference) = 0.553, 95%CI(confidence interval) = 0.418–0.687, $I^2$ = 65.5% $P$<0.05). And this relationship also exists at each sampling time point (NfL specimens were collected on admission: SMD:0.48,95%CI:0.24–0.73; Samples were collected 24 hours after CA: SMD:0.60,95%CI:0.32–0.88;Specimens were obtained 48 hours after CA: SMD:0.51, 95%CI:0.18–0.85;Specimens were obtained 72 hours after CA: SMD:0.59, 95% CI:0.38–0.81).

## Conclusion

NfL may play a potential neuroprognostication role in postcardiac arrest patients with spontaneous circulation, regardless of when the sample was collected after CA.

## Introduction

Out-of-hospital cardiac arrest is still a major cause of death and disability that threatens human health, with an average global incidence among adults of 55 per 100,000 person-years [1]. Although the proportion of spontaneous circulation recovery has greatly increased [2]. owing to the great progress in early initiation of cardiopulmonary resuscitation (CPR), the wide use of automated external defibrillators, and standardized advanced resuscitation, the survival rate of return of spontaneous circulation (ROSC) and neurological function have not been significantly improved [3].

After cardiac arrest, human brain tissue stops due to ischemia and hypoxia, which leads to the massive consumption of aerobic metabolic energy of neurons, and the dysfunction of ion transporters on the cell membrane, resulting in cell edema and neuron damage. As the most abundant protein in the axon of neurons, nerve filament is a key part of the axon scaffold, which plays a role in resisting external pressure, determining axon diameter and regulating transmission speed [4]. The neural filament is composed of core light chain, peripheral protein, heavy chain and middle chain. Damaged neurons release their contents into the cerebrospinal fluid and blood. Filament light chain protein is also released as one of the components of the nerve filament. At present, light chain protein of neural filament is also widely used in the research of degenerative diseases of nervous system. In recent years, there have been attempts to use light chain protein of neural filament to evaluate the prognosis of patients with cardiac arrest. In addition, a systematic review indicated that the NSE threshold for predicting adverse outcomes varied greatly, which may be related to the non-brain-derived origin of NSE, while the predictive value of NfL was higher than that of other biomarkers. However, only two studies in this meta-analysis measured NfL [5]. Another meta-analysis also confirmed that NfL had the best predictive value, but this meta-analysis did not conduct subgroup analysis of the included studies and could not prove whether the predictive value of NfL was affected by other factors [6].

Herein, we performed a meta-analysis to assess the prognostic value of NfL in cardiac arrest patients. In addition, subgroup analysis was performed on sample collection time, neurological function assessment time, study design, acceptance of TTM, specimen determination method, and the presence of neurological disease in patients to assess the influence of these factors on the predictive value of serum NfL.

## Methods

The review was conducted in accordance with the Preferred Reporting Items for Systematic Reviews and Meta-Analyses (PRISMA) statement [7]. This study extracted data from previously published observational studies, so it was not necessary to have the study approved by an ethics committee or institutional review board.

Two investigators (Y Fu and JT Zhang) independently conducted an electronic literature (published before March 12, 2022) search using Embase, Web of Science, Medline, Cochrane, Scopus, ClinicalKey, CINAHL, and PubMed. The publication language was not limited. During our search process, the following key words were used: "respiratory cardiac arrest," "cardiac arrest," "cardio-pulmonary resuscitation," "return of spontaneous circulation," "heart arrest," "asystole," "asystoles," "cardiopulmonary arrest," "neurofilament light chain," "neurofilament," "neurofilament protein L," "neurofilament protein light" "NF-L polypeptide," "NEFL protein," "NEFL polypeptide," "NF-L protein," and "neurofilament light polypeptide."

After the retrieval was completed, two investigators independently assessed the titles, abstracts, and full texts of identified articles to determine whether they met research criteria. A panel of experts was referred to if there was any disagreement in screening. Only studies meeting the following criteria were included: (1) out-of-hospital cardiac arrest patient, (2) age over 16 years old, and (3) NfL specimens were blood specimens. Studies were excluded if they were (1) case reports, meta-analysis or review articles, letters, or editorials, (2) duplicate articles, (3) studies without sufficient data to allow for the extraction of NfL expression levels, (4) animal model or cell line research, (5) studies that did not use cerebral performance category score (CPC) to evaluate outcomes, or (6) considered irrelevant when two independent reviewers identified the titles and abstracts of literature.

Data extraction was performed independently by two researchers (Y Fu and XY Fan), including the following information from each study: basic study information (first author and year of publication), study population characteristics (e.g., individual characteristics, sample size), prognostic evaluation method, number of cases in each prognosis group, time of sample collection, method of NFL testing, type of research, whether the patients had neurological disease and whether or not the study targeted temperature management (TTM) or the mean or median NfL for each prognostic group. We converted quartiles, medians, or maximum and minimum values to means and standard deviations using online calculation tools (https://smcgrath.shinyapps.io/estmeansd/). The calculation method was to convert asymmetric data into symmetric data through the method of Box–Cox transformation and quantile estimation and calculate the mean and SD of the transformed data using the method proposed by Wan et al. [8] and Luo et al. [9] These formulas are widely recognized and used in other meta-analyses [6, 10, 11]. There were inconsistencies that were resolved by a third researcher.

The neurological outcomes were classified into good and poor groups based on CPC scores, we defined the group with a good prognosis as patients with a CPC score of 1–2, that is, patients with a good neurological outcome(GNO), and patients with a CPC3-5 score as patients with a poor neurological prognosis(PNO). The results we were interested in were whether nfl could predict the prognosis of patients with cardiac arrest. Secondary outcomes were whether the time of specimen collection, the time of assessment, the type of study, whether TTM was received, the method of specimen determination, and the presence of neurological disease in patients had any influence on the predictive value of NfL. The QUADAS-2 scale of diagnostic test accuracy analysis in Review Manager software was used to assess the risk of bias for all included studies, mainly evaluating four aspects—case selection, trial to be evaluated, gold standard, and case process and progress—with three possible answers to the relevant questions in each part: yes, no, and uncertain [12]. The corresponding risk offset

grade was low risk, high risk, and uncertain risk. Likewise, the quality evaluation was done independently by two researchers (Y Fu and ZG Chen).

The association between NfL level and neurological outcome was assessed by the standardized mean differences (SMDs) and 95% confidence intervals (CIs) between the patients with GNO and PNO using a random-effects model. Subgroup analysis was conducted according to sample collection time, time to assess neurological function, study design, whether TTM was received, the method of specimen determination, and the presence of neurological disease in patients. The heterogeneity of results across trials was assessed using the $I^2$ statistic, which describes the percentage of total variation across studies that is attributable to heterogeneity rather than to chance. $I^2$ values of 25%, 50%, and 75% correspond to cut-off points for low, moderate, and high degrees of heterogeneity. The pooled effect was calculated using the random-effects model when the $I^2$ value was >75%. Begg's and Egger's tests were used to assess potential publication bias. All statistical analyses were conducted using Review Manager software version 5.4.1 (Cochrane Collaboration) and Stata software version 12.0 (StataCorp, College Station, Texas). A $P$ value <0.05 was considered statistically significant.

## Results

### 1. Characteristics and quality of included studies and subjects

A total of 1209 articles were found through electronic databases. After 134 duplicate articles were deleted, 1075 articles were selected by title screening; after 1088 were excluded according to the exclusion criteria, 67 related articles were selected for abstract screening; 37 articles were abandoned through abstract screening, and 30 articles were retrieved for further analysis. Six studies were finally included in the meta-analysis [13–18] (Fig 1). The characteristics of the included studies are presented in Table 1. In addition, the baseline characteristics and the assay methods detecting NfL concentration are presented in S1 Table in S1 File

### 2. Meta-analysis

**2.1 Association between NfL level and neurological outcomes.** A total of 6 studies were included to assess the level of NfL in the neurologically good and poor prognosis groups, with a total of 1360 subjects. Meta-analysis was performed on the NfL levels of the GNO and the PNO groups at each time point of each study, and it was found that the NfL level in the PNO group was significantly higher than that in the GNO group (SMD = 0.553, 95% CI = 0.418–0.687 $I^2$ = 65.5% $P$<0.05; Fig 2), and the ROC curve suggests that nfl has high specificity and sensitivity in predicting the prognosis of cardiac arrest patients(Fig 3).

**2.2 Correlation of neurological outcomes with NfL level at different sampling time points.** We divided the sampling times in the six studies into admission, 24 hours, 48 hours, and 72 hours after CA. In the admission: SMD:0.48,95%CI:0.24–0.73; Samples were collected 24 hours after CA: SMD:0.60,95%CI:0.32–0.88;Specimens were obtained 48 hours after CA: SMD:0.51, 95%CI:0.18–0.85;Specimens were obtained 72 hours after CA: SMD:0.59, 95% CI:0.38–0.81. The NfL level of the PNO group was higher than that of the GNO group at each measurement time; that is, the NfL had predictive value at different time points. In the admission ($I^2$ = 0%, $P$>0.05) and 72-hour ($I^2$ = 29.0%, $P$>0.05) groups, heterogeneity was low, while the other two groups had high heterogeneity (24-hour group: $I^2$ = 72.5%, $P$<0.05; 48-hour group: $I^2$ = 81.6%; $P$<0.05; Fig 4).

**2.3 Effect of time of neurological function on the predictive value of NfL.** A subgroup analysis of the time of neurological assessment found that the NfL had no neurological assessment ability at 3 months after CA (SMD = 1.18, 95%CI = 0.57–1.79; Fig 5).

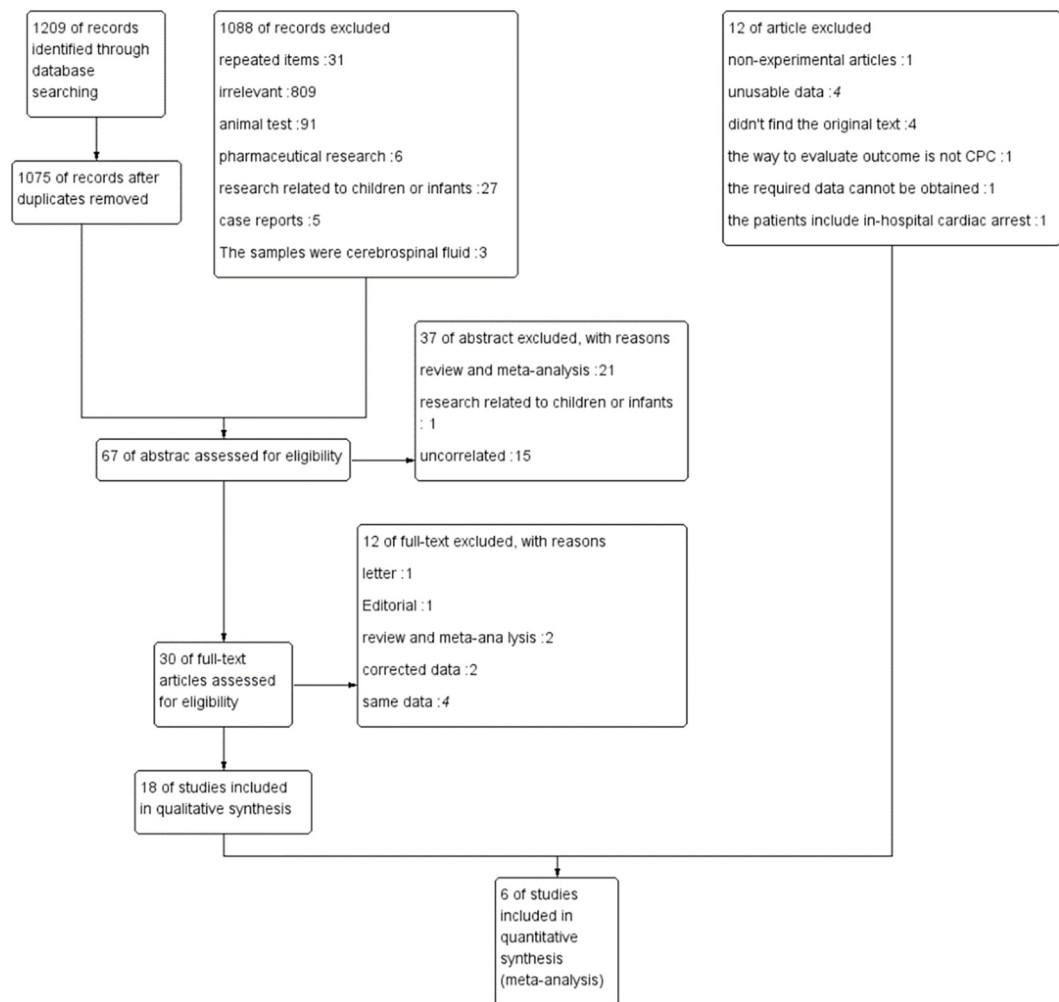

**Fig 1. Flowchart showing selection of the meta-analysis studies.**

**Table 1. The characteristics of the included studies.**

| study | Publication time | Study sites | Study design | Time of sample collection | Sample size (n) | GNO (n) | Time to assess neurological function |
|---|---|---|---|---|---|---|---|
| Hunziker, Sabina [13] | 2021 | Switzerland | prospective observational study | At admission | 164 | 66 | at hospital discharge |
| Moseby-Knappe, M [14] | 2019 | Multi-center study | randomized controlled trial (RCT) | At 24, 48, and 72 hours after CA | 717 | 351 | 6 months after CA |
| Pouplet, C [15] | 2022 | France | RCT | At 48 hours after CA | 59 | 26 | 3 months after CA |
| Raphael Wurm [16] | 2022 | Vienna | prospective observational study | At 48 hours after CA | 70 | 21 | 6 months after CA |
| Wihersaari, L. [17] | 2021 | Multi-center study | RCT | At admission, 24, 48, and 72 hours after CA | 112 | 73 | 6 months after CA |
| Wihersaari, L. [18] | 2022 | Multi-center study | prospective observational study | At 24,48 hours after CA | 248 | 128 | 12 months after CA |

The risk of bias assessment of the included studies is shown in S2 Table in S1 File. Of the six studies included in this study, three studies were rated as low-quality studies using the quality scoring system defined in the Methods section [15, 16, 18] (S1a, S1b Fig in S1 File).

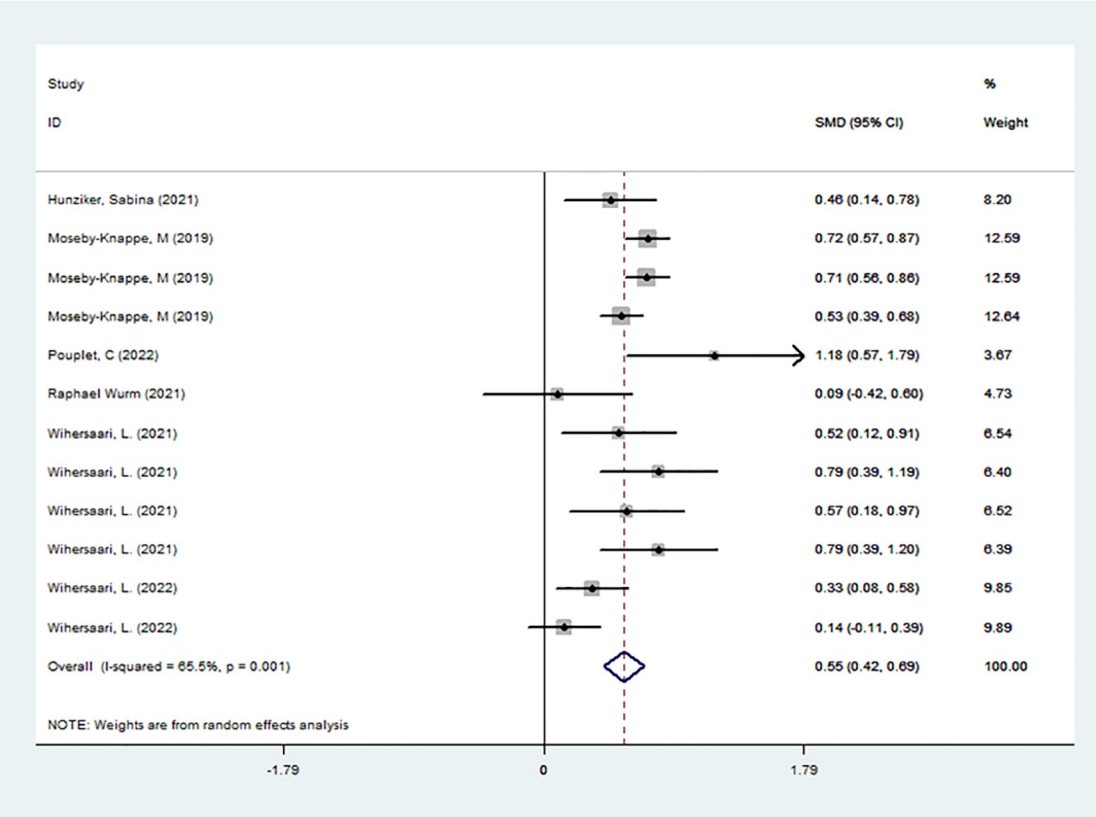

**Fig 2. Data analysis: Statistical analysis of the prognosis of each study at different sampling times.**

**2.4 Subgroup analysis.** *2.4.1 Analysis by study type or instrument to measure the NfL as subgroups.* Study type had no effect on NfL assessment of neurological function in CA patients (S2 Fig in S1 File). Among the included studies, one study measured NfL levels with the Ella microfluidic platform and found that the choice of measurement method had no effect on the predictive power of NFL (S3 Fig in S1 File).

*2.4.2 Effect of TTM treatment on the prognosis of CA patients assessed by NfL.* Regardless of whether the patients received TTM treatment, the NfL level in the poor prognosis group was higher than that in the good prognosis group (subgroup 0: only some of the patients in the study received TTM; subgroup 1: all patients in the study were treated with TTM; S4 Fig in S1 File).

*2.4.3 Subgroup analysis of included patients with or without neurological disease.* In the subgroup analysis with and without neurological diseases, NfL was found to have a predictive effect on the prognosis of CA patients. However, in the group with neurological diseases (subgroup 1), there was moderate heterogeneity, which was related to the few studies in this group (S5 Fig in S1 File).

## 3. Meta-regression analysis

We conducted meta-regression analysis on the age, gender and PCR time of each study population, and found no statistical significance (S6 Fig in S1 File).

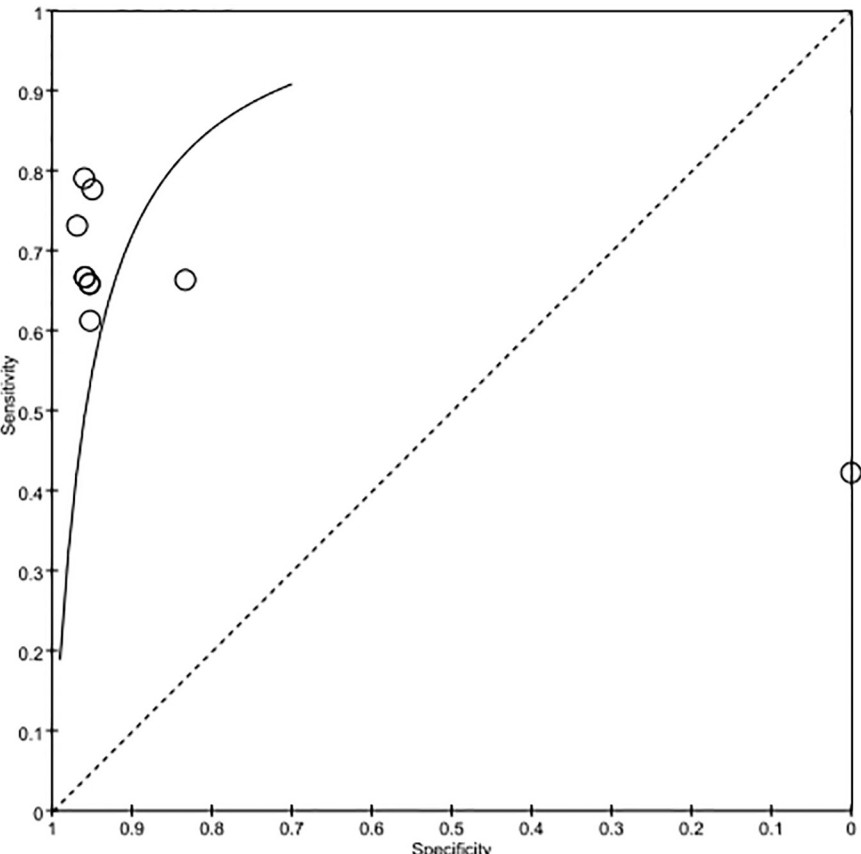

**Fig 3. ROC: Receiver operating characteristic curve.**

## 4. Sensitivity analysis and publication bias

There was no significant heterogeneity in the sensitivity analysis (S7 Fig in S1 File). Begg's funnel plot showed no obvious asymmetry in publication bias. In Egger's test, *P*>0.05 (S8a, S8b Fig in S1 File).

## Discussion

In this meta-analysis, we assessed differences in serum NfL concentrations between the good and poor prognosis groups from six studies to assess the potential of NfL as a biomarker. NfL levels were increased in the poor prognosis group compared with the good prognosis group, with high levels of NfL may indicate a poor prognosis in patients with out-of-hospital cardiac arrest.

After cardiac arrest, the brain tissue is damaged by hypoxic–ischemic injury and ischemia–reperfusion injury, resulting in neuronal damage and apoptosis. At present, the measures predicting neurological outcomes after cardiac arrest include neuroelectrophysiology (electroencephalography, somatosensory evoked potential, etc.), imaging examination (CT, cranial magnetic resonance, optic nerve sheath diameter measurement), physical examination, and biological markers (NSE, tau protein, etc.). However, neuroelectrophysiology requires a high level of expertise to interpret, imaging examinations are expensive, it is not suitable for monitoring, and physical examinations are greatly affected by sedative drugs. Biomarkers are not

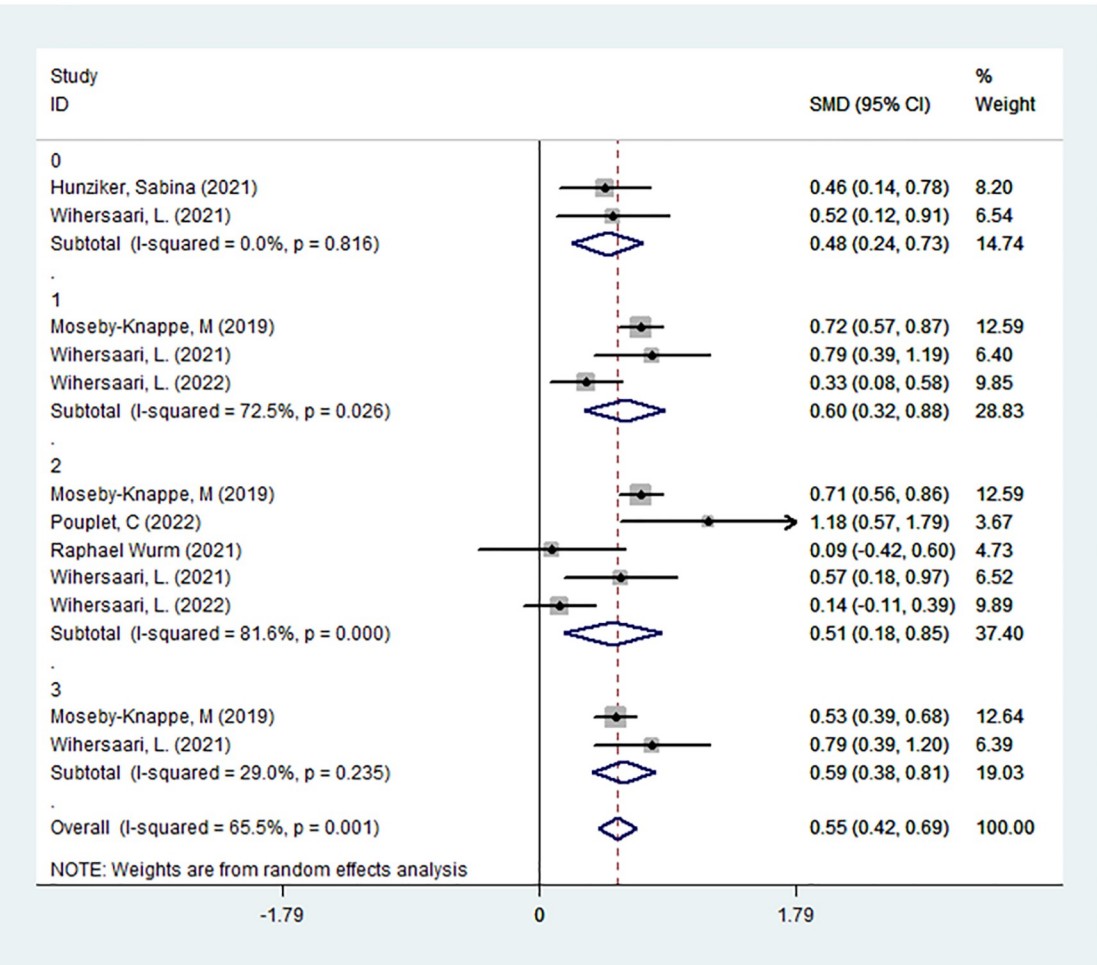

**Fig 4. Subgroup analysis was performed at different sampling times.** 0: NfL specimens were collected on admission; 1: Samples were collected 24 hours after CA; 2: Specimens were obtained 48 hours after CA; 3: Specimens were obtained 72 hours after CA.

affected by drugs or hypothermia. The ideal biomarker should be expressed specifically and exclusively. Conventional biomarkers, such as NSE, showed low sensitivity when their specificity was assessed very high, and remained low sensitivity when their specificity was reduced. This underscores the importance of using methods such as the NfL with high discriminant accuracy. Of the mature myelinated axons, neurofilaments are the most abundant protein. Damage and physiological turnover of neurons in the central nervous system (CNS) result in the release of nerve filaments. As a structural protein, NfL is released into the cerebrospinal fluid and blood when neurons are damaged or apoptotic. As a neuron-specific marker of neurological injury, elevated NfL levels can be found in a variety of conditions involving neuroaxonal injury in both the central and peripheral nervous system. In neurodegenerative diseases, NfL could serve as both a prognostic marker of decline but also an efficacy biomarker of experimental therapies [19]. In a meta-analysis of Alzheimer's disease, frontotemporal and amyotrophic lateral sclerosis, plasma NfL levels were elevated in patients compared to controls with utility in differentiating neurodegenerative conditions from non-neurodegenerative mimics [20]. Different from the invasive lumbar puncture, which increases the risk of infection in patients, serological study is relatively convenient and easy to operate, as well as having high

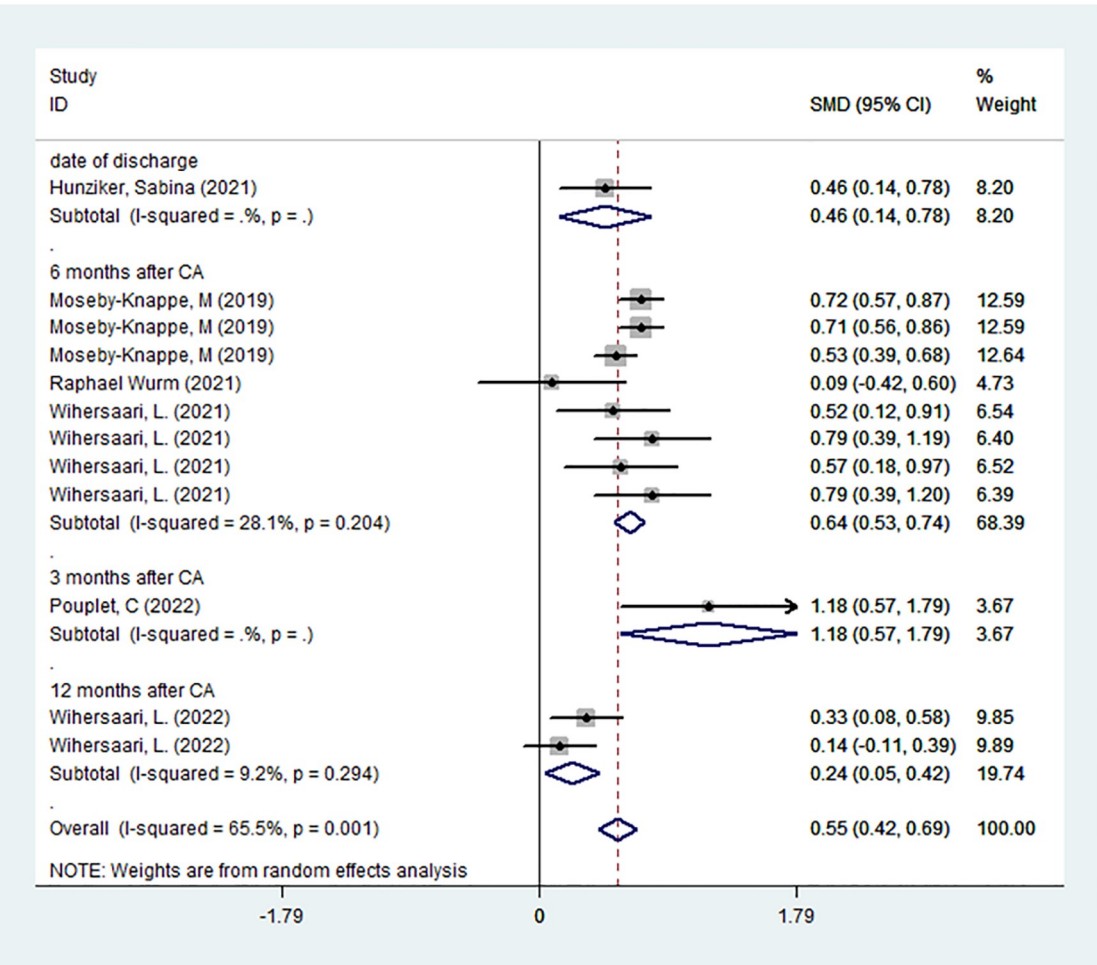

**Fig 5. NfL influence of time to assess neurological function on NfL prediction of neurological function.**

specificity in patients with cardiac arrest. Compared with NSE and tau protein [6, 14, 21], NFL has higher predictive value and is comparable to other conventional predictive methods (head CT, EEG, SSEP, and tau). Compared with pupillary or corneal reflexes, NfL levels predict poorer neurological outcomes with greater sensitivity at the same specificity [14].

We analyzed predictive levels of NfL at various time points and found that NfL levels at admission and 72 hours after CA were highly prognostic, but only one study has completed the collection of NfL at admission. Therefore, although our results are satisfactory, more studies are needed to confirm one idea. Serum NfL also had good predictive value at 24 and 48 hours, with high heterogeneity, while its predictive value at 24 and 48 hours was equivalent to that at 72 hours among COMACARE populations [17]. Similarly, a recent study found that the predictive value 48 hours after CA was better than at admission [22]. Therefore, we have reason to believe that the prediction ability of serum NfL at 24 and 48 hours after CA is better after increasing the sample size.

In subgroup analyses based on the time point of assessment, serum NfL concentration was found to be predictive of neurological function at discharge, 6 months, and 12 months after CA, but the neuroprognostication value at 3 months is unclear. However, only one study evaluated neurological function at 3 months after CA, in which the level of NFL had no

neuroprognostication value at each time point of assessment. Our study may have enrolled more severe patients, creating heterogeneity with other studies (registration website: https://clinicaltrials.gov/ct2/show/NCT02555254). At present, there are few studies on the evaluation of neurological function in CA patients at three months, which is insufficient to support our conclusions.

The predictive value of NfL was not affected by TTM treatment, and a higher level of NfL predicted poorer neurological outcomes. Heterogeneity was low in general but increased after the combination of the two subgroups (the subgroup in which all patients in the study were treated with TTM and the subgroup in which only some of the patients in the study received TTM), while the lack of control groups without TTM may make the results more complicated. We found that the methods of NfL study had no effect on its predictive value, which calls for further well-designed, perspective, multicenter trials to confirm. Furthermore, we also performed subgroup analysis of study type and presence of neurological diseases and found that these two subgroups had no effect on the predictive effect of NfL.

There are several limitations to our meta-analysis. First, some of the included studies were of poor quality, but the risk of bias is that the study did not set a threshold, which had no effect on the study. Second, the included studies were heterogeneous. Third, there was a lack of control groups without TTM. Fourth, a large proportion of the studies were conducted in patients of European ethnicity. Therefore, more studies are required to assess the level of NfL in a variety of ethnic groups.

## Conclusions

Serum NfL levels can be used to assess neurological function in patients with cardiac arrest. Higher levels of NfL suggest poorer prognosis, especially at admission and 72 hours after cardiac arrest. However, NfL level is not predictive of neurological function at 3 months after CA, which requires more studies and further analyses.

## Supporting information

**S1 Checklist.**
(DOC)

**S1 File. Contains figures and tables from the article.**
(DOCX)

**S2 File. Study's minimal data.**
(DOCX)

## Acknowledgments

We thank LetPub (www.letpub.com) for its linguistic assistance during the preparation of this manuscript.

## Author Contributions

**Conceptualization:** Yu Fu, Hui Li.

**Data curation:** Yu Fu, Xiao-Tian Fan, Ding-Ding Zhang, Hao Jiang.

**Formal analysis:** Yu Fu.

**Methodology:** Yu Fu, Hui Li, Zhi-Guo Chen.

**Software:** Yu Fu.

**Validation:** Yu Fu, Ran Zhang.

**Writing – original draft:** Yu Fu.

**Writing – review & editing:** Zhi-Guo Chen, Jiang-Tao Zhang.

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
