## [Decision Letter · Decision Letter 0]

1 Dec 2022

PONE-D-22-27382

Neuroprognostication value of serum neurofilament light chain for out-of-hospital cardiac arrest: a systematic review and Meta-analysis

PLOS ONE

Dear Dr. Chen,

Thank you for submitting your manuscript to PLOS ONE. After careful consideration, we feel that it has merit but does not fully meet PLOS ONE’s publication criteria as it currently stands. Therefore, we invite you to submit a revised version of the manuscript that addresses the points raised during the review process.

We have excellent input from three reviewers, two of which (2 and 3) highlight pertinent and fundamental challenges in this manuscript. Should you be able to extensively revise the manuscript according to these comments it may be reconsidered for publication. Should you find this unfeasible, I suggest retracting the paper from further consideration. 

We look forward to receiving your revised manuscript.

Kind regards,

Jussi Olli Tapani Sipilä, M.D., Ph.D., B.Soc.Sci.

Academic Editor

PLOS ONE

2. We note that this manuscript is a systematic review or meta-analysis; our author guidelines therefore require that you use PRISMA guidance to help improve reporting quality of this type of study. Please upload copies of the completed PRISMA checklist as Supporting Information with a file name “PRISMA checklist”.

“No.The funders had no role in study design, data collection and analysis, decision to publish, or preparation of the manuscript.”

Reviewers' comments:

Reviewer's Responses to Questions

**Comments to the Author**

1. Is the manuscript technically sound, and do the data support the conclusions?

Reviewer #1: Yes

Reviewer #2: Partly

Reviewer #3: Partly

2. Has the statistical analysis been performed appropriately and rigorously? 

Reviewer #1: Yes

Reviewer #2: No

Reviewer #3: Yes

3. Have the authors made all data underlying the findings in their manuscript fully available?

Reviewer #1: Yes

Reviewer #2: No

Reviewer #3: Yes

4. Is the manuscript presented in an intelligible fashion and written in standard English?

Reviewer #1: Yes

Reviewer #2: Yes

Reviewer #3: Yes

5. Review Comments to the Author

Reviewer #1: the present is a meta-analysis evaluating the impact of serum neurofilament light chain for out-of-hospital

cardiac arrest

Abstract, methods>prognosis analysis is not clear and should be better detailed

Methods (both in abstract and in results)>it should be clearly stated if these results derive from univariate or multivariate analysis

Abstract>conclusion.The authors wrote "regardeless on then samples were collected". This was not reported in the results, so they should either report in the results or remove from the conclusion

methods>exclusion of patients younger than 16 years should be discussed

methhods>conversion of unit of measures into quertiles, means etc was performed throuhg online calculator (please add reference)

methods should be divided into paragraphs

methods>primary end point should be defined

methods>it is not clear if the correlation derived (for eacht study) from multivariate analysis or not. This is a crucial point

methods>subgroup analuysis for rct vs. non rct should be added

discussion; the results are strong, although they should be discussed cautiosuly

Reviewer #2: Review comments PONE-D-22-27382

Fu and colleagues conducted a systematic review and meta-analysis on the neuro-prognostic value of neurofilament light in out of hospital cardiac arrest patients. While neurofilament light is indeed a promising biomarker for neuro-prognostication in the patient population, and likely holds future utility in clinical practice, the novelty/utility of this review is unclear to me. Two previous systematic reviews including Nf-L have recent been published (see below) and the authors have not indicated in their manuscript how./why it is novel. Further, the data presentation is relatively simplistic and doesn’t provide scientists or clinicians with metrics that they can use to advance their knowledge or inform their clinical practice. For these major reasons and my comments below I am skeptical of what impact this paper would provide to the field and hope that the authors find my comments (while critical) to be helpful for improving their manuscript.

Comments

#1 The authors state that “no systematic review has been conducted to assess [NfL’s] diagnostic performance”, but this is quite frankly false. The authors even cite a recent paper that HAS conducted a systematic review on NfL (Ref 16: Hoiland et al., 2022, JAMA Neurol), and fail to cite another that has also (Sandroni et al., 2020, Intensive Care Med, 46:1803-1851). Continuously, more and more scientists are refraining from the use of statements of primacy to avoid erroneous assertions such as this one. I would implore the authors to also refrain from statements of primacy herein, and in future work. The above quoted statement is used in Lines 17-18 of the abstract, but the authors should thoroughly scan the document to ensure others have not slipped through my review:

#2 In the introduction the rationale presented for the use of NfL and why this systematic review is necessary is quite rudimentary, and there is no description of what NfL as a biomarker is a proxy for. I suggest the authors expand their introduction to inform readers as to what NfL actually is, what it tells us about brain injury, and for what reasons it may be better to use than other markers. Further, why this review is needed in lieu of the two recent ones noted above is also warranted.

#3 Why were data converted to mean and standard deviation when these biomarker data are (in my experience) almost never normally distributed? I am not a statistician but I believe that this presents a major flaw from an analytical perspective.

#4 Figure 1 shows 6 included studies. The prior systematic review and meta-analysis by Hoiland et al., 2022, JAMA Neurol included 10. Could the authors please clarify the discrepancy and if studies were missed? My guess is they may not have been specific to OHCA, but I am unsure.

I have scanned the Hoiland et al., 2022, JAMA Neurol paper and found the following papers they included but are not present here. It would be informative to clarify why they were excluded:

1. Adler et al. 2021; PMID: 34328545

2. Disanto et al. 2019; PMID: 31375414

3. Hoiland e; t al. 2021PMID: 34287000

4. Huesgen et al. 2021; PMID: 34223394

5. Moseby-Knappe et al. 2021; PMID: 34417831

6. Rana et al. 2013; PMID: 23287695

#5 Figure 3 needs some explanation. What does the 0, 1, 2, 3, & 4 denote? Different sampling times? Please add the necessary details to interpret this figure within the figure legend.

#6 I have reservations about how this paper can be utilized by other researchers and clinicals to inform either their experiments or clinical practice, respectively, which should be a major consideration for a meta-analysis of this nature. To provide utility I would presume that metrics such as the summary statistics for NfL in the poor and good outcome groups (e.g. median [IQR]), cut-offs for prognosticating good vs poor outcome (e.g. receiver-operator curve characteristics), and the related sensitivity and specificities of prognostic cut-offs should be presented. If the authors specifically did not want to include these metrics rationale as to why should be presented.

#7 Line 174: Please provide additional details as to what NfL is. Structural in what way and in what tissue?

Reviewer #3: I thank you for the opportunity to comment on this meta-analysis and review that concerns neurofilament light as a prognostic tool after cardiac arrest. There is still a reduced number of studies on NfL but it seems to be superior over all the other biomarkers. Biomarkers, as they are cheap, non-invasive and reliable, is a rising topic in multimodal neuroprognostication.

Overall, the meta-analysis itself is suitable and language is fluent. There is, however, some major problems and things to be improved. The greatest issue is that this review does not offer much additional information to readers. The second issue is that there is some inconsistency and errors in definition of accuracy of biomarkers, and in their comparison. The third one is the chapter Discussion that deals with this topic too generally.

I have some major and minor comments on this manuscript.

Abstract:

Major comments:

In the beginning there is a sentence “ no systematic review has been conducted to assess its diagnostic performance.” I have to disagree with that, please look at this review: Hoiland RL, et al. Neurologic Prognostication After Cardiac Arrest Using Brain Biomarkers: A Systematic Review and Meta-analysis. JAMA Neurol. 2022 Apr 1;79(4):390-398

I am asking if this review gives any additional information? There are only six studies in this review, do you think that the whole question statement is correct? I think that maybe you should find something more, something to add to this JAMA review. Also, there is no demonstration any statistical comparison to other diagnostic tools (biomarkers, imaging or neurophysiological). More on, there is problems in methodology – in determination of the prognostic accuracy of NfL and other tests, and how to compare them. Now this review is too superficial and offers only a little important additional information. Taken as a whole, my suggestion is to make the corrections I am asking for, and find some additional information to the Hoiland et al´s work – for example comparing NfL to other prognostic tools?

Minor comments:

There is abbreviations in the abstract (CPC, SDM, CI) – please explain them when used first time and if necessary, you should shorten the abstract accordingly.

The main text:

Introduction

Major comments:

The sentence starting from line 44 “It has been suggested that a higher NfL level may predict a higher likelihood of death, providing more favorable prognostic performance compared with NSE . “ is innaccurate. You are right that higher NfL means higher likelihood of poor outcome but that is not an argument that makes it better than NSE. NfL discriminated patients with poor outcome from those with good outcome in those studies, demonstrated as significantly higher AUROCs. In prognostication after CA, the sensitivity of biomarker to detect those patients with potentially good outcome (with low biomarker values) is a very important topic. We can set the cut-off very high and get 100% of specificity for all studied biomarkers and tests, and only well discriminating tests still have ability to find some patients with good outcome also. In addition, when using less accurate test, the high specificity means that there is many false negative results and the clinical value of that test is poor.

I strongly recommend to modify the Introduction chapter, so readers get a better understanding about neuroprognostication (there is not a need for a very long text, however) and why some biomarkers/tests are more suitable for clinical use than others. Please give some more references.

Minor comments:

Line 41: The recovery of neurological function is one of the most important factors in determining the 42 prognosis in patients with ROSC.

Comment: You should give some references for that sentence.

Line 42: prognosis in patients with ROSC. Biomarkers such as neuron-specific enolase (NSE) and S100-β 43 protein are recommended due to their convenience

Comment: Also this sentence needs a reference(s). I would recommend you to underline that prognostication after CA should be multimodal and there is also some problems using NSE and S100B (eg. false positive results in NSE due to haemolysis and small-cell lung cancer). We can´t give too promising appearance for using only biomarkers in prognostication.

Line 43: …Recently, a novel biomarker, neurofilament light chain (NfL), has been used for neuroprognostication in patients with ROSC.

Comment: I think this sentence is incorrect – NfL has only been studied post hoc, not used as a prognostic tool. You should rewrite this sentence and give references.

Methods

There is searches from many databases. That´s good. The criteria for meta-analysis are well described and suitable.

Minor comments:

Starting from line 57: During our search process, the following key words were used: “respiratory cardiac arrest,” “cardiac arrest,” “cardiopulmonary resuscitation,” “return of spontaneous circulation,” “heart arrest,” “asystole,” “asystoles,” “cardiopulmonary arrest,” “neurofilament light chain,” “neurofilament,” “neurofilament protein L,” “neurofilament protein light” “NF-L polypeptide,” “NEFL protein,” “NEFL polypeptide,” “NF-L 61 protein,” and “neurofilament light polypeptide.”

Comment: In previous sentence you mentioned the used keywords. I would like to ask if you tested the search with abbreviations, eg CPR, CA, ASY, VF, OHCA? I am not sure if that would give any more studies. I ask also if you searched with keywords: out-of-hospital cardiac arrest and ventricular fibrillation?

The NfL values are presented as means with standard deviations. The distribution of NfL is highly sqewed and accordingly I recommend rather to use the median values with interquartile ranges. Regarding to the prognostic ability, I suggest to use AUROCs, as mentioned elsewhere.

Results

This is a well-made chapter and additional data is showed in supplementary materials.

Meta-analysis

Major comments:

Starting from line 128: “The NfL level of the PNO group was higher than that of the GNO group at each measurement time; that is, the NfL had predictive value at different time points.”

This should be written in more precise way. As I mentioned earlier, the higher values, of course, is an indication of likelihood of poor outcome. However, it is highly recommended to use AUROCs as a definition of discriminative ability, and further, to demonstrate the prognostic value.

Minor comments:

in the Figures 2 and 4 I can´t find a mention of sampling times – please demonstrate those in the pictures. Now it is a bit confused.

Discussion

Major comments:

There is the same problems than earlier: the information is too general and you do not actually show any references to prove your text. Also, you said that NSE and S100B had a poor specificity but high sensitivity – that is too imprecisely said. Sensitivity and specificity are absolutely affected by the selected cut-off value. NSE has demonstrated to have worse sensitivity than NfL in selected high specificities, which is used when a low amount of false positive results is required, as it is in predicting poor outcome.

Please rewrite this part of the review more precisely regarding the definition of accuracy of the diagnostic tests and add the required references.

6. PLOS authors have the option to publish the peer review history of their article (what does this mean?). If published, this will include your full peer review and any attached files.

Reviewer #1: **Yes: **Fabrizio D'Ascenzo

Reviewer #2: No

Reviewer #3: **Yes: **Lauri Wihersaari

---

## [Author Response · Author response to Decision Letter 0]

31 Mar 2023

Dear editor:

Thank you for your review of our articles and your valuable comments. Here are our responses to reviewers' comments.

Requirements for journals:

1.We have modified according to the writing requirements of PLOS ONE.

2.Our article was written in accordance with the PRISMA checklist, and its corresponding content is submitted again.

3.We have made changes to our financial disclosures and completed the changes in “cover letter”. Our article was funded by a Science and Technology Project of Hebei Province (approval number :20377764D) in our province.

Funders help us complete the revision of the article and provide financial support.

4.The minimum basic data set of the study has been submitted to the website of this journal as a supporting information file.

5.The title of the supporting information file has been added at the end of the manuscript.

Answers to the following questions:

1.Is the manuscript technically sound, and do the data support the conclusions?

Our article was written in accordance with PRISMA's checklist, and some content may not be submitted in time, which has been added in this revision.

2.Has the statistical analysis been performed appropriately and rigorously?

When collecting data, we found that the available data provided in the included studies were in the form of median and standard deviation, and the original data could not be found, so we adopted a data conversion formula for data processing. Our research has been completed, and this formula has been proved reliable and adopted by many research institutes.(Hoiland et al., 2022, JAMA neurology; Li et al., 2022, The Lancet Respiratory medicine )

3.Have the authors made all data underlying the findings in their manuscript fully available? 

The minimum basic data set of the study has been submitted to the website of this journal as a supporting information file.

4.Is the manuscript presented in an intelligible fashion and written in standard English?

The language of our article is English.

Reply to Reviewer #1：

1.Abstract, methods>prognosis analysis: 

The standard mean difference and 95% confidence interval were calculated using the random-effects model or fixed-effects model to assess the association between one variable factor NfL level and the outcome of CA patients. 

2.Methods (both in abstract and in results)>it should be clearly stated if these results derive from univariate or multivariate analysis

Assess the association between one variable factor NfL level and the outcome of CA patients. 

3.Abstract>conclusion.The authors wrote "regardeless on then samples were collected". This was not reported in the results, so they should either report in the results or remove from the conclusion.

"regardeless on then samples were collected" that means 24 hours, 48 hours and 72 hours after cardiac arrest. Statistics for three points in time have been added to the results section.

4.methods>exclusion of patients younger than 16 years should be discussed.

The subjects included in the study are adults. Due to the differences in regional population differentiation, people under the age of 16 are excluded.

5.methhods>conversion of unit of measures into quertiles, means etc was performed throuhg online calculator (please add reference).

Hoiland et al., JAMA neurology 2022; Li et al., The Lancet Respiratory medicine 2022; Su et al., Medicina (Kaunas, Lithuania) 2021.

6.methods>primary end point should be defined.

The results we were interested in were whether nfl could predict the prognosis of patients with cardiac arrest.

7.methods>it is not clear if the correlation derived (for eacht study) from multivariate analysis or not. This is a crucial point.

The data for statistical analysis were all from similar populations, and the data were processed according to time and detection methods.

8.methods>subgroup analuysis for rct vs. non rct should be added.

We have analyzed the ability of study types to predict outcomes in NfL in a subgroup analysis.(Figure S2).

9.discussion; the results are strong, although they should be discussed cautiosuly.

NfL levels were increased in the poor prognosis group compared with the good prognosis group, with high levels of NfL may indicate a poor prognosis in patients with out-of-hospital cardiac arrest.

Reply to Reviewer #2:

1. Although meta-analysis has confirmed its predictive value, it has not conducted a more detailed analysis of its research. We conducted a meta-analysis to evaluate the relationship between serum NfL level and neurological prognosis in patients with spontaneous circulation recovery after CA, and subgroup analysis of the included studies was conducted to evaluate the influence of other factors on the predictive value of NfL.

2.We have filled in the missing parts proposed in the article, and compared the two articles mentioned by the reviewer.

3.The NfL data included in the study were not normally distributed data, but the mean and standard deviation were required for statistical analysis, so we converted the data, and the formula we used was proved to be reliable by other studies and has been used by many studies.( Hoiland et al., 2022, JAMA neurology; Li et al., 2022, The Lancet Respiratory medicine )

4.Exclude the reasons for the following articles：

Adler et al. 2021; PMID: 34328545: It did not use cerebral performance category score (CPC) to evaluate outcomes.

Disanto et al. 2019; PMID: 31375414: Fourteen patients were included in this paper, but 15 cardiac arrests were recorded, and the timing of specimen collection was not stated.

Hoiland e; t al. 2021PMID: 34287000: The study did not provide criteria or timing for assessing nervous system function.

Huesgen et al. 2021; PMID: 34223394: The study included a very different population than other studies. 

 Moseby-Knappe et al. 2021; PMID: 34417831: This study was incorporated into our analysis

Rana et al. 2013; PMID: 23287695: It did not use cerebral performance category score (CPC) to evaluate outcomes.

5.Figure 3：0: NfL specimens were collected on admission; 1: Samples were collected 24 hours after CA; 2: Specimens were obtained 48 hours after CA; 3: Specimens were obtained 72 hours after CA.

6. We added the NfL roc curve to the article to provide more intuitive evidence

7. At the original line 174, we added the following:

Of the mature myelinated axons, neurofilaments are the most abundant protein. Damage and physiological turnover of neurons in the central nervous system (CNS) result in the release of nerve filaments. As a structural protein, NfL is released into the cerebrospinal fluid and blood when neurons are damaged or apoptotic. As a neuron-specific marker of neurological injury, elevated NfL levels can be found in a variety of conditions involving neuroaxonal injury in both the central and peripheral nervous system. In neurodegenerative diseases, NfL could serve as both a prognostic marker of decline but also an efficacy biomarker of experimental therapies. In a meta-analysis of Alzheimer’s disease, frontotemporal and amyotrophic lateral sclerosis, plasma NfL levels were elevated in patients compared to controls with utility in differentiating neurodegenerative conditions from non-neurodegenerative mimics. 

Reply to Reviewer #3:

Abstract:-- Major comments:

We made the following modifications: “Although meta-analysis has confirmed its predictive value, it has not conducted a more detailed analysis of its research. We conducted a meta-analysis to evaluate the relationship between serum NfL level and neurological prognosis in patients with spontaneous circulation recovery after CA, and subgroup analysis of the included studies was conducted to evaluate the influence of other factors on the predictive value of NfL.”

Minor comments:

CPC: cerebral performance category score

SMD: standardized mean difference

CI:confidence interval

The main text:

Introduction--Major comments:

“In recent years, there have been attempts to use light chain protein of neural filament to evaluate the prognosis of patients with cardiac arrest. In addition, a systematic review indicated that the NSE threshold for predicting adverse outcomes varied greatly, which may be related to the non-brain-derived origin of NSE, while the predictive value of NfL was higher than that of other biomarkers. However, only two studies in this meta-analysis measured NfL5. Another meta-analysis also confirmed that NfL had the best predictive value, but this meta-analysis did not conduct subgroup analysis of the included studies and could not prove whether the predictive value of NfL was affected by other factors”

Minor comments:

The three sentences mentioned by the Reviewer #3 have been removed, and the preamble has been substantially altered.

Methods--Major comments:

The abbreviations we use include CA, CPR, ROSC, NfL. But we didn't use out-of-hospital cardiac arrest and ventricular fibrillation.We added the ROC curve to the NfL's predictive power.

Results:

Meta-analysis--Major comments:

In the admission: SMD:0.48,95%CI:0.24-0.73; Samples were collected 24 hours after CA: SMD:0.60,95%CI:0.32-0.88;Specimens were obtained 48 hours after CA: SMD:0.51, 95%CI:0.18-0.85;Specimens were obtained 72 hours after CA: SMD:0.59, 95%CI:0.38-0.81

We were unable to obtain serum NFL values for every patient included in the study, so the ROC curve could not be completed.

Minor comments:

Figure 2 shows the inclusion of results from each study at each point in time

Figure 4: 0: NfL specimens were collected on admission; 1: Samples were collected 24 hours after CA; 2: Specimens were obtained 48 hours after CA; 3: Specimens were obtained 72 hours after CA.

Discussion--Major comments:

The discussion section has been modified and references added.

6. PLOS authors have the option to publish the peer review history of their article (what does this mean?). If published, this will include your full peer review and any attached files.

Yes

 Yours sincerely,

 Fu Yu

---

## [Decision Letter · Decision Letter 1]

10 May 2023

PONE-D-22-27382R1血清神经丝轻链对院外心经停的神经预测后价值：系统评价和荟萃分析PLOS ONE

Dear Dr. Chen,

Thank you for submitting your manuscript to PLOS ONE. After careful consideration, we feel that it has merit but still does not fully meet PLOS ONE’s publication criteria as it currently stands. Therefore, we invite you to submit a further revised version of the manuscript that addresses the points raised during the review process.

We look forward to receiving your revised manuscript.

Kind regards,

Jussi Olli Tapani Sipilä, M.D., Ph.D., B.Soc.Sci.

Academic Editor

PLOS ONE

Reviewers' comments:

Reviewer's Responses to Questions

**Comments to the Author**

1. If the authors have adequately addressed your comments raised in a previous round of review and you feel that this manuscript is now acceptable for publication, you may indicate that here to bypass the “Comments to the Author” section, enter your conflict of interest statement in the “Confidential to Editor” section, and submit your "Accept" recommendation.

Reviewer #1: (No Response)

Reviewer #3: All comments have been addressed

2. Is the manuscript technically sound, and do the data support the conclusions?

Reviewer #1: Yes

Reviewer #3: No

3. Has the statistical analysis been performed appropriately and rigorously? 

Reviewer #1: Yes

Reviewer #3: I Don't Know

4. Have the authors made all data underlying the findings in their manuscript fully available?

Reviewer #1: Yes

Reviewer #3: Yes

5. Is the manuscript presented in an intelligible fashion and written in standard English?

Reviewer #1: Yes

Reviewer #3: Yes

6. Review Comments to the Author

Reviewer #1: The present is an interesting meta-analysis, aiming to evaluate impact of Neurofilament light chain on neurological function after cardiac arrest.

Some issues need to be addressed

1) major issues. In the abstract and in the introduction authors stated that others meta analysis have been planned, although this should be better described. in particular it should be added which new informations will be provived by the present paper

2) definition of good and poor prognosis should be better described in the methods

3) In the analysis it is not clear when authors used random effect

4) subgroup analysis should be performed for rct vs. observational

5) meta-regression for age, gender and time of rcp should be performed

Reviewer #3: The authors have answered my questions satisfactorily. However, there are some other issues in the discussion that I have to disagree.

In lines 204-205 You state: “Traditional biomarkers, such as NSE and S100β, are not affected by drugs or hypothermia and have high sensitivity but poor specificity”. Those markers are more specific than sensitive, and overall, the specificity for each biomarker can be assessed very high (99-100%) BUT when the prognostic accuracy (the ability to discriminate according to outcome) is poor, the sensitivity remains low. This underlines the importance of using methods with high discriminative accuracy (high AUROCs) as NfL is. I suggest rewriting this sentence.

In lines 221-224 You say that admission and 72 h NfL has higher prognostic accuracy than at 24-48 h. This is not in line with our findings and with those of Moseby-Knappe et al. In the COMACARE population, the admission NfL had only moderate accuracy, whereas it was excellent at 24, 48 and 72 h similarly to in the TTM analysis. In a recent analysis, NfL had slightly better (and not inferior) ability at 48 h than at 12 h and admission sample had only satisfactory prognostic ability (Levin H, Lybeck A, Frigyesi A, Arctaedius I, Thorgeirsdóttir B, Annborn M, Moseby-Knappe M, Nielsen N, Cronberg T, Ashton NJ, Zetterberg H, Blennow K, Friberg H, Mattsson-Carlgren N. Plasma neurofilament light is a predictor of neurological outcome 12 h after cardiac arrest. Crit Care. 2023 Feb 24;27(1):74 ). The kinetic of NfL seems to be that it rises 12-24 h after CA and the prognostic accuracy remains high between 24 and 72 h. You analyzed admission NfL only in a one study. Even if Your analysis suggests those results (maybe affected by heterogeneity) I suggest modifying this chapter and do not suggest the superiority of admission NfL because the studies do not support this. Moreover, You should modify the conclusion accordingly.

Lines 225-230: I am asking if the outcome definition at 3 mth offers any benefit compared to 6-mth outcome. Most studies define outcome at 6 months. Maybe “the best CPC or mRS” would offer some additional benefit in studying the prognostic accuracy of neuronal biomarkers?

7. PLOS authors have the option to publish the peer review history of their article (what does this mean?). If published, this will include your full peer review and any attached files.

Reviewer #1: **Yes: **Fabrizio D'Ascenzo

Reviewer #3: No

---

## [Author Response · Author response to Decision Letter 1]

20 Jun 2023

Dear editor:

Thank you for your review and decision regarding our manuscript, titled “Neuroprognostication Value of Serum Neurofilament Light Chain for Out-of-hospital Cardiac Arrest: A Systematic Review and Meta-Analysis”. We appreciate the reviewers’ and editor’s insightful and helpful comments on our manuscript. 

We sincerely thank all reviewers for their insightful comments. We have provided a point-by-point response to the reviewers’ comments. The changes in the revised manuscript are highlighted.

Response to the reviewer’s comments

1. If the authors have adequately addressed your comments raised in a previous round of review and you feel that this manuscript is now acceptable for publication, you may indicate that here to bypass the “Comments to the Author” section, enter your conflict of interest statement in the “Confidential to Editor” section, and submit your "Accept" recommendation.

We have amended and recovered the issues raised.

2. Is the manuscript technically sound, and do the data support the conclusions?

Response: After rigorous selection of articles, statistical analysis was carried out on the data included in the study, and appropriate conclusions were drawn according to the data analysis results.

3. Has the statistical analysis been performed appropriately and rigorously?\\

Data extraction, selection and application of statistical analysis methods were completed by two researchers independently, and the results were consistent. Moreover, the statistical methods we used were introduced in detail in the methodological section.

4. Have the authors made all data underlying the findings in their manuscript fully available?

We put the main data in the article, and the rest of the data is uploaded as supporting information.

5. Is the manuscript presented in an intelligible fashion and written in standard English?

The language used in our article is English and has been revised.

6. Review Comments to the Author

Our paper is a statistical analysis of the data from the original research, and there is no research ethics issue, and the paper will not be published twice.

Responses to reviewer #1：

1) major issues. In the abstract and in the introduction authors stated that others meta analysis have been planned, although this should be better described. in particular it should be added which new informations will be provived by the present paper

We appreciate the reviewer’s comments. We agree with the reviewer’s concerns about the purpose of the article is not clear. We added the statement of the research purpose in the abstract and the preface respectively.

In the background of the abstract, we have added "We conducted a meta-analysis to evaluate the relationship between serum NfL level and neurological prognosis in patients with spontaneous circulation recovery after CA, and subgroup analysis was conducted according to sample collection time, time to assess neurological function, study design, whether TTM was received, the method of specimen determination, and the presence of neurological disease in patients. To analyze the influence of these factors on the predictive value of serum NfL." (Line18-23)

At the end of the preface, we have added "Herein, we performed a meta-analysis to assess the prognostic value of NfL in cardiac arrest patients. In addition, subgroup analysis was performed on sample collection time, neurological function assessment time, study design, acceptance of TTM, specimen determination method, and the presence of neurological disease in patients to assess the influence of these factors on the predictive value of serum NfL."(Line65-69)

2) definition of good and poor prognosis should be better described in the methods

The neurological outcomes were classified into good and poor groups based on CPC scores, we defined the group with a good prognosis as patients with a CPC score of 1-2, that is, patients with a good neurological outcome(GNO), and patients with a CPC3-5 score as patients with a poor neurological prognosis(PNO).(Line104-107)

3) In the analysis it is not clear when authors used random effect

It has been proposed in line 123-125 that when I2>75%, using the random effects model.

4) subgroup analysis should be performed for rct vs. Observational

The study type was statistically analyzed as a subgroup and showed no effect on the predictive value of serum NfL.(Line181-182)

5) meta-regression for age, gender and time of rcp should be performed

We conducted meta-regression analysis on the age, gender and PCR time of each study population, and found no statistical significance.(Line195-197)

Responses to reviewer #3

In lines 204-205 You state: “Traditional biomarkers, such as NSE and S100β, are not affected by drugs or hypothermia and have high sensitivity but poor specificity”. Those markers are more specific than sensitive, and overall, the specificity for each biomarker can be assessed very high (99-100%) BUT when the prognostic accuracy (the ability to discriminate according to outcome) is poor, the sensitivity remains low. This underlines the importance of using methods with high discriminative accuracy (high AUROCs) as NfL is. I suggest rewriting this sentence.

We rewrote the sentence: “Biomarkers are not affected by drugs or hypothermia. The ideal biomarker should be expressed specifically and exclusively. Conventional biomarkers, such as NSE, showed low sensitivity when their specificity was assessed very high, and remained low sensitivity when their specificity was reduced. This underscores the importance of using methods such as the NfL with high discriminant accuracy.”

In lines 221-224 You say that admission and 72 h NfL has higher prognostic accuracy than at 24-48 h. This is not in line with our findings and with those of Moseby-Knappe et al. In the COMACARE population, the admission NfL had only moderate accuracy, whereas it was excellent at 24, 48 and 72 h similarly to in the TTM analysis. In a recent analysis, NfL had slightly better (and not inferior) ability at 48 h than at 12 h and admission sample had only satisfactory prognostic ability (Levin H, Lybeck A, Frigyesi A, Arctaedius I, Thorgeirsdóttir B, Annborn M, Moseby-Knappe M, Nielsen N, Cronberg T, Ashton NJ, Zetterberg H, Blennow K, Friberg H, Mattsson-Carlgren N. Plasma neurofilament light is a predictor of neurological outcome 12 h after cardiac arrest. Crit Care. 2023 Feb 24;27(1):74 ). The kinetic of NfL seems to be that it rises 12-24 h after CA and the prognostic accuracy remains high between 24 and 72 h. You analyzed admission NfL only in a one study. Even if Your analysis suggests those results (maybe affected by heterogeneity) I suggest modifying this chapter and do not suggest the superiority of admission NfL because the studies do not support this. Moreover, You should modify the conclusion accordingly.

We revised the conclusion: “We analyzed predictive levels of NfL at various time points and found that NfL levels at admission and 72 hours after CA were highly prognostic, but only one study has completed the collection of NfL at admission. Therefore, although our results are satisfactory, more studies are needed to confirm one idea. Serum NfL also had good predictive value at 24 and 48 hours, with high heterogeneity, while its predictive value at 24 and 48 hours was equivalent to that at 72 hours among COMACARE populations17. Similarly, a recent study found that the predictive value 48 hours after CA was better than at admission22. Therefore, we have reason to believe that the prediction ability of serum NfL at 24 and 48 hours after CA is better after increasing the sample size.”

Lines 225-230: I am asking if the outcome definition at 3 mth offers any benefit compared to 6-mth outcome. Most studies define outcome at 6 months. Maybe “the best CPC or mRS” would offer some additional benefit in studying the prognostic accuracy of neuronal biomarkers?

The neurological function evaluation of cerebrovascular diseases is generally conducted 3 months after the occurrence of cerebrovascular accidents. Therefore, there is a certain basis for evaluating the neurological function prognosis of patients 3 months after CA. Therefore, we believe that we can increase the research of neurological function evaluation 3 months after CA in the future.

We appreciate your re-evaluation of our revised manuscript. We would be glad to respond to any further questions and comments that you may have.

Sincerely,

Zhi-Guo Chen

---

## [Editor Report · Decision Letter 2]

27 Jun 2023

PONE-D-22-27382R2Neuroprognostication Value of Serum Neurofilament Light Chain for Out-of-hospital Cardiac Arrest: A Systematic Review and Meta-AnalysisPLOS ONE

Dear Dr. Chen, 

Thank you for submitting your manuscript to PLOS ONE. After careful consideration, we feel that it has merit but does not fully meet PLOS ONE’s publication criteria as it currently stands. Therefore, we invite you to submit a revised version of the manuscript that addresses the points raised during the review process.

Please submit your revised manuscript by Aug 11 2023 11:59PM.  If you will need more time than this to complete your revisions, please reply to this message or contact the journal office at plosone@plos.org. Please include the following items when submitting your revised manuscript:A rebuttal letter that responds to each point raised by the academic editor and reviewer(s). You should upload this letter as a separate file labeled 'Response to Reviewers'.A marked-up copy of your manuscript that highlights changes made to the original version. You should upload this as a separate file labeled 'Revised Manuscript with Track Changes'.An unmarked version of your revised paper without tracked changes. You should upload this as a separate file labeled 'Manuscript'.If applicable, we recommend that you deposit your laboratory protocols in protocols.io to enhance the reproducibility of your results. Protocols.io assigns your protocol its own identifier (DOI) so that it can be cited independently in the future. For instructions see: https://journals.plos.org/plosone/s/submission-guidelines#loc-laboratory-protocols. Additionally, PLOS ONE offers an option for publishing peer-reviewed Lab Protocol articles, which describe protocols hosted on protocols.io. Read more information on sharing protocols at https://plos.org/protocols?utm_medium=editorial-email&utm_source=authorletters&utm_campaign=protocols.

We look forward to receiving your revised manuscript.

Kind regards,

Jussi Olli Tapani Sipilä, M.D., Ph.D., B.Soc.Sci.

Academic Editor

PLOS ONE

Journal Requirements:

**Additional Editor Comments:**

"The neurological function evaluation of cerebrovascular diseases is generally conducted 3 months after the occurrence of cerebrovascular accidents. Therefore, there is a certain basis for evaluating the neurological function prognosis of patients 3 months after CA. Therefore, we believe that we can increase the research of neurological function evaluation 3 months after CA in the future."

This is not an acceptable logic. Stroke is a very different entity compared to post-resuscitation. Especially as your own results show that the only available data does not support neuroprognostication at 3 months there is no basis to endorse it here. Moreover, instead of blindly suggesting "more research is needed" on this, as is routine, you could analyze if any are needed (PMID: 37338877).

The paper was not sent to the external reviewers at this stage

---

## [Author Response · Author response to Decision Letter 2]

5 Jul 2023

Dear editor:

Thank you for your review and decision regarding our manuscript, titled “Neuroprognostication Value of Serum Neurofilament Light Chain for Out-of-hospital Cardiac Arrest: A Systematic Review and Meta-Analysis”. We appreciate the reviewers’ and editor’s insightful and helpful comments on our manuscript. 

We sincerely thank all reviewers for their insightful comments. We have provided a point-by-point response to the reviewers’ comments. The changes in the revised manuscript are highlighted.

None of the references we cited have been retracted.

Response to the reviewer’s comments

"The neurological function evaluation of cerebrovascular diseases is generally conducted 3 months after the occurrence of cerebrovascular accidents. Therefore, there is a certain basis for evaluating the neurological function prognosis of patients 3 months after CA. Therefore, we believe that we can increase the research of neurological function evaluation 3 months after CA in the future."

This is not an acceptable logic. Stroke is a very different entity compared to post-resuscitation. Especially as your own results show that the only available data does not support neuroprognostication at 3 months there is no basis to endorse it here. Moreover, instead of blindly suggesting "more research is needed" on this, as is routine, you could analyze if any are needed (PMID: 37338877).

In subgroup analyses based on the time point of assessment, serum NfL concentration was found to be predictive of neurological function at discharge, 6 months, and 12 months after CA, but the neuroprognostication value at 3 months is unclear. However, only one study evaluated neurological function at 3 months after CA, in which the level of NFL had no neuroprognostication value at each time point of assessment. Our study may have enrolled more severe patients, creating heterogeneity with other studies (registration website: https://clinicaltrials.gov/ct2/show/NCT02555254). At present, there are few studies on the evaluation of neurological function in CA patients at three months, which is insufficient to support our conclusions.

We appreciate your re-evaluation of our revised manuscript. We would be glad to respond to any further questions and comments that you may have.

Sincerely,

Zhi-Guo Chen

---

## [Decision Letter · Decision Letter 3]

14 Aug 2023

Neuroprognostication Value of Serum Neurofilament Light Chain for Out-of-hospital Cardiac Arrest: A Systematic Review and Meta-Analysis

PONE-D-22-27382R3

Dear Dr. Chen,

We’re pleased to inform you that your manuscript has been judged scientifically suitable for publication and will be formally accepted for publication once it meets all outstanding technical requirements.

Kind regards,

Jussi Olli Tapani Sipilä, M.D., Ph.D., B.Soc.Sci.

Academic Editor

PLOS ONE

Additional Editor Comments (optional):

Reviewers' comments:

Reviewer's Responses to Questions

**Comments to the Author**

1. If the authors have adequately addressed your comments raised in a previous round of review and you feel that this manuscript is now acceptable for publication, you may indicate that here to bypass the “Comments to the Author” section, enter your conflict of interest statement in the “Confidential to Editor” section, and submit your "Accept" recommendation.

Reviewer #1: All comments have been addressed

Reviewer #3: All comments have been addressed

2. Is the manuscript technically sound, and do the data support the conclusions?

Reviewer #1: (No Response)

Reviewer #3: Yes

3. Has the statistical analysis been performed appropriately and rigorously? 

Reviewer #1: (No Response)

Reviewer #3: Yes

4. Have the authors made all data underlying the findings in their manuscript fully available?

Reviewer #1: (No Response)

Reviewer #3: Yes

5. Is the manuscript presented in an intelligible fashion and written in standard English?

Reviewer #1: (No Response)

Reviewer #3: Yes

6. Review Comments to the Author

Reviewer #1: (No Response)

Reviewer #3: Thank you for the revised manuscript. Although I could not find exact answers to my questions/comments, the manuscript is acceptable after this modifications. I have no additional comments.

7. PLOS authors have the option to publish the peer review history of their article (what does this mean?). If published, this will include your full peer review and any attached files.

Reviewer #1: **Yes: **Fabrizio Dascenzo

Reviewer #3: **Yes: **Lauri Wihersaari

---

## [Editor Report · Acceptance letter]

8 Sep 2023

PONE-D-22-27382R3 

Neuroprognostication Value of Serum Neurofilament Light Chain for Out-of-hospital Cardiac Arrest: A Systematic Review and Meta-Analysis 

Dear Dr. Chen:

I'm pleased to inform you that your manuscript has been deemed suitable for publication in PLOS ONE. Congratulations! Your manuscript is now with our production department. 

Kind regards, 

on behalf of

Dr. Jussi Olli Tapani Sipilä 

Academic Editor

PLOS ONE